

# Measured and modelled air quality related effects of a noise barrier near a busy highway

Sami D. Harni[1], Lasse Johansson[1], Jarkko V. Niemi[2], Ville Silvonen[3], Juan Andrés Casquero-Vera[4,5], Anu Kousa[2], Krista Luoma[1], Viet Le[1], David Brus[1], Konstantinos Doulgeris[1], Topi Rönkkö[3], Hanna E. Manninen[2], Tuukka Petäjä[4], and Hilkka Timonen[1]

[1]Atmospheric Composition Research, Finnish Meteorological Institute, PL 503, FIN-00101 Helsinki, Finland
[2]Helsinki Region Environmental Services Authority HSY, Helsinki, Finland
[3]Aerosol Physics Laboratory, Physics Unit, Faculty of Engineering and Natural Sciences Tampere University, Korkeakoulunkatu 3, 33720 Tampere, Finland
[4]Institute for Atmospheric and Earth System Research, Faculty of Science, University of Helsinki, Helsinki 00014, Finland
[5]Andalusian Institute for Earth System Research, IISTA-CEAMA, University of Granada, Junta de Andalucía, Granada, 18006, Spain

*Correspondence to*: Sami Harni (sami.harni@fmi.fi)

**Abstract.** A three-month air quality measurement campaign was conducted in spring 2023 near a busy highway in Espoo, Finland. The measurement site featured a high (6.5 m) noise barrier built adjacent to the highway. Additionally, there was a gap in the noise barrier at the selected measurement site, providing an opportunity to study the air quality impacts of the noise barrier. Several air quality measurement devices were installed behind the noise barrier and in the gap at distances of 10, 20 and 40 m from the side of the highway. Additionally, 15 passive samplers were deployed to monitor $NO_2$ concentrations across the study area, mobile measurements were conducted using the ATMo-Lab mobile laboratory on the highway and concurrent flights with drones equipped with AQ monitors were performed along the highway.

The effects of the noise barrier on $PM_{10}$, $PM_{2.5}$, lung deposited surface area (LDSA), particle number concentration (PNC), $NO_2$, and black carbon (BC) were quantified based on analysed measurement data. Furthermore, the measurements were compared with simulated pollutant concentrations from a local scale Gaussian air quality model (Enfuser) with a nearby obstacle detection and concentration reduction method incorporated in the model to address the effects of the noise barrier in the study.

The noise barrier was found to effectively reduce pollutant concentrations behind the barrier. The most significant reductions were observed closest to the highway. The greatest reductions were observed for $PM_{10}$ (mostly road dust) while gaseous concentrations, such as $NO_2$, exhibited less pronounced decreases.

## 1 Introduction

Traffic significantly influences air quality near highways through both exhaust emissions (Hilker et al., 2019; Sofowote et al., 2018) and wearing products originating from pavement, tyres and brakes as well as NaCl from winter salting (Denby et al.,



2016; Kupiainen et al., 2016; Pirjola et al., 2010; Sofowote et al., 2018; Vouitsis et al., 2023). These emissions pose risks to both human health and climate (Baensch-Baltruschat et al., 2020; WHO, 2021). The composition and size of particulate emissions from traffic vary significantly depending on the source. Combustion processes produce vast amounts of black carbon

and organic-containing nanoparticles (Rönkkö & Timonen, 2019). The non-exhaust particles from traffic tend to be in the coarse mode size range and contain a mix of organics, metallic trace elements and NaCl from road salting (Denby et al., 2016; Vouitsis et al., 2023). Advancements in exhaust after-treatment technologies have been highly effective in reducing particulate emissions from traffic (Gren et al., 2021). These systems, combined with the increasing electrification of vehicles have already led to a significant reduction in exhaust-related emissions. Current EU legislations on particle emissions from traffic exhaust

are so stringent that, in some jurisdictions, non-exhaust particles now represent a larger share of traffic-related particulate matter emissions (Fussell et al., 2022). In Europe, the upcoming Euro 7 legislation will also regulate tyre and brake wear emissions.

The worldwide interest in reducing atmospheric pollutants is of great importance. This is reflected by WHO air quality guidelines that include pollutants such as $PM_{2.5}$, $PM_{10}$, $O_3$, $NO_2$, $SO_2$ and CO. However, ultrafine particle number concentration

(PNC) and elemental carbon (EC) are also included in good practice statements and systematic measurements of them are encouraged WHO, 2021. Also, significantly stricter limits for various pollutants are going to be implemented with the new EU air quality directive with an obligation to measure ultrafine particle (UFP) and black carbon (BC) concentrations (European Air quality Directive, 2024).

Distance from the road has been found to significantly affect pollutant concentrations, with notable decreasing gradients

observed for NO and $NO_2$ as distance increases from roads (Thoma et al., 2008). Pollutant concentrations perpendicular to highways have been found to follow logarithmic gradients, with the steepest decrease occurring near the road (Zheng et al., 2022). Similarly shaped gradients of pollutant concentration on the side of the highway have been observed also as a function of time after being emitted on the road (Kangasniemi et al., 2019). The gradients for particles are found to be more pronounced for larger particles (Zheng et al., 2022). In an earlier study, noise barriers have been found to reduce $NO_x$ concentrations by

23 % at 5 behind the noise barrier (Tezel-Oguz et al., 2023). Notably lower concentrations have been observed for traffic-related ultrafine particle, BC, CO and $NO_2$ concentrations behind the noise barrier compared to areas with no noise barrier even at 300 m behind the noise barrier by Baldauf et al. (2016). The steepness of the pollutant gradient has been found to be strongly dependent on the flow dynamics (Enroth et al., 2016). This would suggest that noise barriers strongly affect the dispersion of pollutants on the roadside. The beneficial effects of noise barriers on air quality have also been reported by Li et

al. (2021).

Several modelling approaches have been developed to meet the demand for accurate and timely information on urban AQ (Johansson et al., 2022; Rolstad Denby et al., 2020). The spatial variability of pollutant concentrations within an urban environment is pronounced, which poses a challenge for air quality modelling systems. To facilitate operational timely production the urban scale models are mostly Gaussian dispersion models. However, the proper modelling of fluid dynamics

in a complex environment would need more sophisticated tools such as Lagrangian particle simulations or Large Eddy



Simulations (LES) (Hellsten et al., 2021). Unfortunately, the computational cost of using such sophisticated tools makes the adoption of such approaches infeasible.

The objectives of this study can be summarized to: what is the effect of the noise barriers on the pollutant gradients and concentrations on the side of a highly trafficked highway and, how do the modelling results compare to the measurements? More specifically this study aimed to characterise the influence of a noise barrier on air quality near a highway using a sensor type and mobile measurement platforms. Sensors were used to measure $PM_{10}$, $PM_{2.5}$, $NO_2$, BC, LDSA and $NO_2$ concentrations as a function of distance (10, 20 and 40 m) from the highway, both behind a noise barrier and in an open area. PNC was also measured using sensor measurement but only at 20 and 40 m poles. In addition, two condensation particle counters (CPCs) were used to study PNCs at the 20 m distances. Mobile measurements were conducted at the adjacent highway with the mobile laboratory. Additionally, fifteen passive samplers were deployed to monitor $NO_2$ concentrations around the measurement location.

Another aim of this study was to test the capability of a Gaussian urban scale dispersion model to quantify the effects of urban obstacles such as noise barriers while using computationally lightweight methods, which are feasible for operational use. The data included here provides urgently needed information about the influence of noise barriers and can be used to make better annual air quality maps for roadside locations for city planners.



## 2 Methodology

### 2.1 Measurement campaign location and infrastructure

**Figure 1: Measurement infrastructure at the measurement site. The noise barrier has been marked to figure with orange lines. The measurement poles behind the noise barrier (NB10m, NB20m, and NB40m) have been marked with red and the measurement poles in the open area (O10m, O20m, and O40m) have been marked with blue. Passive NO₂ samplers (PAS_N100m, PAS_N37m, PAS_S1m, PAS_S20m, PAS_6m, PAS_NB60m, PAS_NB86m, PAS_O60m and PAS_O86m) have been marked in yellow. There were also passive samplers located in the main measurement poles.  Also, a mobile laboratory is presented in the figure. (© Google Earth 2024)**

Measurements were conducted from March 1ˢᵗ to May 31ˢᵗ of 2023 near a busy highway (60°12'10.7"N, 24°44'14.1"E) that connects two major cities (Helsinki and Turku) in Southern Finland. According to (DigiTraffic, 2024) the daily traffic flow at the highway was approximately 63,000 vehicles/working day with an average speed of 85 km/h. The share of heavy-duty vehicles was approximately 5% during working days (Monday to Friday). The measurement site was located between a warehouse and the highway (Fig. 1). Next to the highway, approximately 7 m from the highway, a 6.5 m high noise barrier



had been built. Additionally, there was a 100 m gap in the noise barrier in front of the warehouse. The area around the measurement poles between the noise barrier and the warehouse was flat with only grass and no obstacles including trees that could have affected the dispersion of pollutants.

## 2.2 Fixed instrumentation at the campaign site

Two rows of measurement poles were installed at distances of 10, 20 and 40 m from the highway (Fig. 1). One row of sensors
was situated behind the noise barrier (poles named NB10m, NB20m, NB40m) while the other row was placed in the open area (O10m, O20m, O40m). The measuring height of the air quality sensors was about 2 m. In addition to the measurement poles, 15 $NO_2$ passive samplers were set around the measurement location. Six of the passive samplers were located at the measurement poles and the other 9 in locations seen in Fig 1.

Following air quality parameters related to street dust and exhaust gases were measured: $PM_{10}$ and $PM_{2.5}$ (particle mass with
diameter < 10 or 2.5µm, Vaisala model AQT530, Note: AQT530 Vaisala measured particle size > 0.6 µm, Petäjä et al. (2021)), $NO_2$ (AQT530, Vaisala; IVL type passive samplers), black carbon (BC; AE51, TSI; ObservAir, DSTech), particle number concentration (PNC, AQ Urban, Pegasor) and lung deposited surface area (LDSA; AQ Urban, Pegasor; Partector, Naneos). Vaisala AQT530 sensor is described by Petäjä et al., (2021), Pegasor AQ Urban by (Kuula et al., 2020), BC sensors by (Cheng & Lin, (2013)) (AE51) and (Caubel et al., (2019) (ObservAir). IVL-type passive sampler for $NO_2$ is described by Ferm (1991)
and Ayers et al. (1998). Two CPCs, Airmodus A30 and Brechtel 1720 were used to measure total particle number concentrations for particles larger than 7 nm at poles O20m and NB20m, respectively. However, these additional measurements fall out of the scope of this work and can be used as data input in future studies. Before and after the highway campaign the air quality sensors were tested in co-location measurements with reference instruments at air quality monitoring stations. In the colocation measurements, the BC sensors were compared to MAAP. The average deviation from the MAAP
was calculated and the measurement data was corrected by multiplying the data with the derived correction factor. A similar process was also used for LDSA sensors (Partector) to make the concentrations to the same level between the instruments. Additionally, Partector sensors were also made comparable to AQ Urban by multiplying the LDSA concentrations by 0.66.

Data coverages of data usable in analysis for the different instruments are presented in the supplemental materials Table S1.
For $NO_2$ only the data from March was used as the authors wanted to be sure that all 6 instruments were functioning properly at the same time, and during April and May the weather got warmer, and the measurement data seemed to include more errors and was therefore left out of analysis. Also, the BC data from April and March was removed from the analysis as the large fast temperature changes between day and night caused artefacts in the data. The temperature-related issues of BC sensors are well described by (Elomaa et al., 2024).




## 2.3 Weather conditions during the measurement campaign

The weather conditions including hourly temperature, boundary layer height, precipitation, and water on the road are presented in Fig 2. During the measurement campaign, the meteorological conditions changed from winter conditions with subzero temperatures (minimum hourly temperature -10°C.) to spring with temperatures up to +20°C. The precipitation (mostly snow) was quite frequent in March, whereas only some precipitation (mostly rain) was observed in April and May. Especially the beginning of April was distinctively dry. The presented weather conditions are a mix of measured and modelled information; the boundary layer height originates from the SILAM chemical transport model (CTM) (Sofiev et al., 2015) and the water column height on the road surface is taken from the closest optical road weather measurement site (DigiTraffic, 2024).

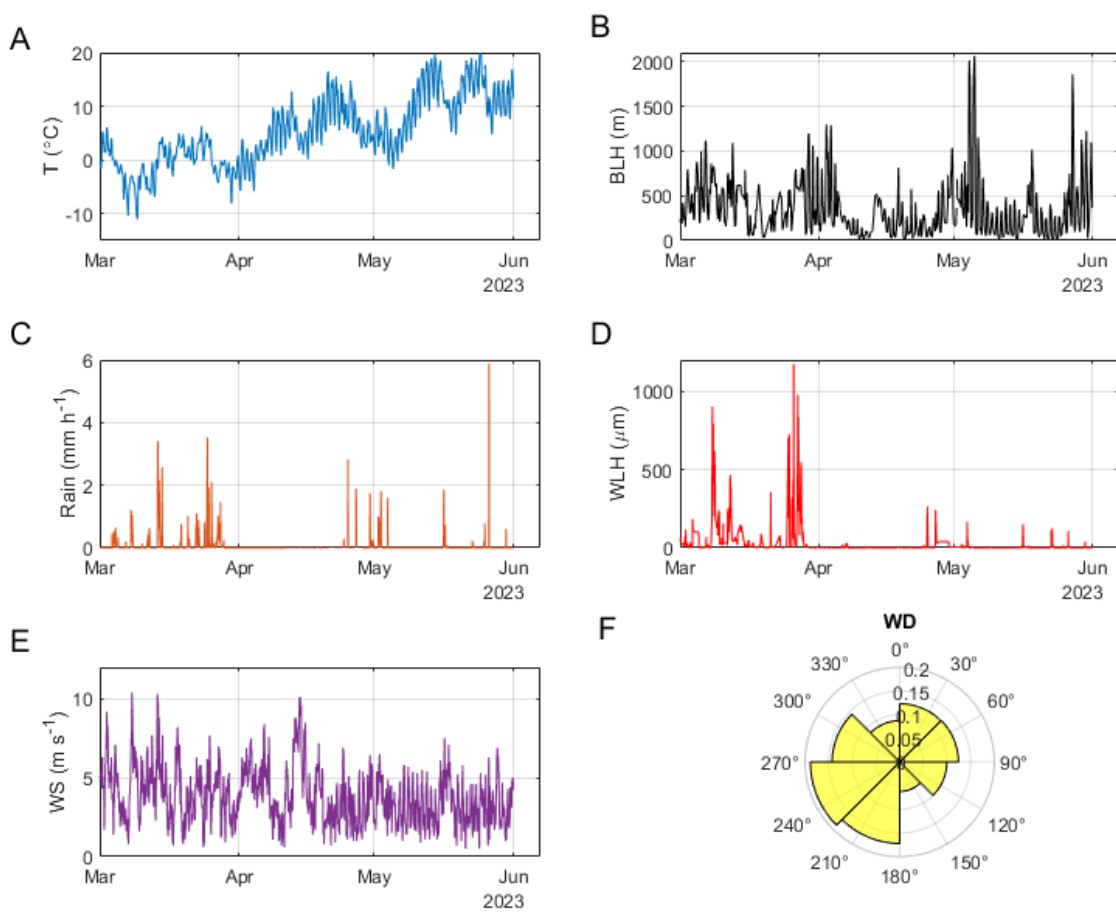

**Figure 2: The weather data including A) Temperature (*T*), B) boundary layer height (BLH), Rain, C) water layer height on road (WLH), E) wind speed (WS) and F) Wind direction (WD) during the measurement period (March 1st to May 31st, 2023).**



## 2.4 Mobile and drone measurements

Mobile measurements on the highway were conducted using the ATMo-Lab mobile laboratory (Lepistö et al., 2023; Rönkkö et al., 2017). The air was sampled from the top of the windshield at a height of ~2.2 m with a flow rate of 22 lpm. An 8 km loop was driven on two measurement days (Monday 20th March and Friday 24th March) between 11 am and 3 pm, resulting in 300 minutes of highway measurement data. Total particle number concentrations were measured with four CPCs with different D50 % cut diameters: 2.5 nm (TSI 3756), 4 nm (TSI 3775), 10 nm (Airmodus A20) and 22 nm (Airmodus A23). The size distribution of particles was measured with an electrical low-pressure impactor (Dekati ELPI+). Additionally, black carbon mass concentration (Magee Scientific AE33), particle chemical composition (Aerodyne Research Inc, soot-particle aerosol mass spectrometer; SP-AMS), $CO_2$-concentration (LI-COR LI-850) and $NO_x$-concentration (Teledyne T-201) were measured. Of the measured components the $NO_2$, LDSA, PNC, and BC are compared to the stationary measurements in this paper by using the concentrations measured with the mobile laboratory as a reference point on the road.

Drone measurements were conducted with a custom-built multicopter built around the Tarot X6 hexacopter platform. The copter had a maximum flight time of 15 min. The maximum take-off weight for the copter was around 11 kg. During flight the GPS locations and altitudes of the copter were recorded using (GPS – Global Positioning System – position fix) The scientific instrumentation was attached to purposebuild modules. The exact description of the drone is given by Brus et al., (2021). The scientific payload of the drone consisted of condensational particle counters (CPCs; model 3007, TSI Corp.; total count in the range from 0.01 to >1.0 µm), the mini cloud droplet analyser (mCDA, Palas GmbH) with a size range of 0.2-17 µm in 256 bins resolution), a micro Aethalometer (model MA200, microAeth), a basic meteorological sensor (Bosch BME280; P, pressure; T, temperature; and RH, relative humidity), and Raspberry 4 microcomputer. The data from the instruments was recorded on the microcomputer with a 1 Hz resolution. The drone measurements are used in this paper for evaluating the vertical dispersion of pollutants.

## 2.4 Modelling framework

Enfuser is a Gaussian local scale air quality model that uses AQ measurement–driven data assimilation (Johansson et al., 2022). Among other areas of interest, Enfuser is currently being used operationally in the Helsinki metropolitan area in Finland. The outputs of the model are hourly average pollutant concentrations (e.g., $PM_{2.5}$, $PM_{10}$, $NO_2$, $O_3$, PNC, BC and LDSA) at a breathing height of 2 m above ground, and this output provided is publicly available via FMI's Open Data portal (FMI, 2024). In addition, hourly updating air quality index (AQI) visualizations based on the model results are available at (HSY, 2024). The key emission sources being modelled include road traffic emissions (also resuspension of particles), residential wood combustion, marine shipping, and local power plants.

Details on the Enfuser modelling system and its input sources in the Helsinki metropolitan area have been presented in (Johansson et al., 2022). In summary, the most essential information sources are as follows:



- HARMONIE numerical weather prediction model (NWP) (Bengtsson et al., 2017) is used as the main source for meteorological input. In addition, the local road weather measurement network gives additional information on e.g., road surface moisture that impacts the modelling of $PM_{10}$.
- The regional background is given by the SILAM CTM model (Sofiev et al., 2015).
- Online AQ measurement results are extracted from the FMI Open data portal. In addition, complementary sources for AQ data have been included to incorporate e.g., AQ sensors that are maintained by local authorities.
- A wide range of heterogeneous GIS inputs is used and assimilated to characterize the modelling area and urban structures within the area. The most notable source in this regard is the OpenStreetMap (OSM).

Enfuser model for the whole Helsinki metropolitan area is used to predict hourly pollutant concentrations from January 1st of 2023 up to 31st of May 2023. During the modelling all available reference quality AQ measurement data is used in the model's data assimilation to adjust source-specific emission release rates and the regional background (Johansson et al, 2022). However, the results of sensors used in this measurement campaign have all been excluded from this set of inputs to facilitate an unbiased comparison against model predictions. In addition to the modelled concentration fields for the Helsinki metropolitan area, the model is used to assess high-resolution concentration predictions with a 4x4 m grid (base resolution 13x13 m) around the measurement campaign area for the duration of the campaign. The modelling duration prior to this can be regarded as a spin-up period for data assimilation methods adopted in the model. Finally, the modelled concentrations are compared against the measured hourly concentration for all sensors. In this comparison, the model is used to predict the concentrations using the exact coordinates and listed measurement heights for all the sensors, as opposed to fetching the model predictions from the raster output with predefined resolution.

### 2.4.1 Modelling the effects of the noise barrier

The Enfuser model uses various geographic information sources to describe the modelling area and its characteristics including buildings and urban objects via OpenStreetMap (Johansson et al., 2022). The vicinity of urban structures and their impact on the dispersion of pollutants can be addressed in a simplified manner considering the limitations of Gaussian dispersion modelling techniques. The approach has been illustrated in Fig. 3. For any given location of interest (referred to as a receptor point, RP) for which model predictions are made, the surrounding area is first scanned to detect urban obstacles. This scanning is done in 10-degree sectors up to 125 m. In case an obstacle is detected the angle of observation, and the height of the object



are logged. In case there are multiple obstacles with varying heights the obstacle with a higher angle of observation takes

priority.

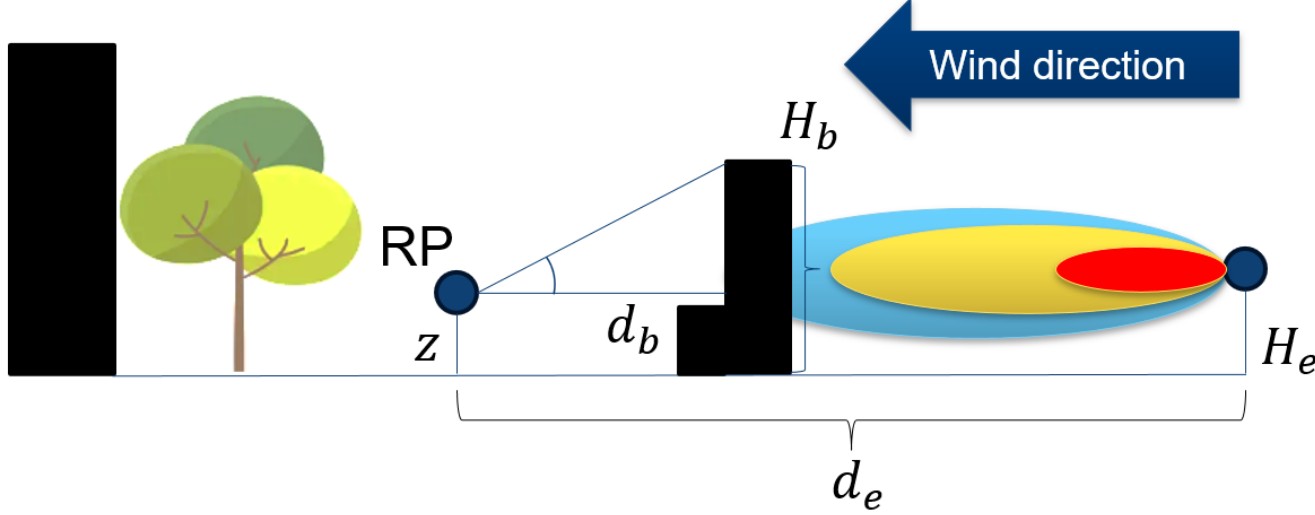

**Figure 3: Illustration of the modelling approach and known distance and height measures for reducing concentrations due to barriers. A singular emission source at a distance of $d_e$ is shown from a receptor point (RP) in which the concentrations are**
**computed. Given the ambient wind direction, the precomputed distances to local barriers or buildings and their heights are fetched. In case an obstacle exists between the emitter and the RP then the shown distance and height measures are used to apply a reduction for modelled Gaussian plume concentrations.**

Let us consider an emission source ($e$) at some location near the RP for which the emission release rate [$\mu gs^{-1}$] for a pollutant

species is known. Depending on the measurement height $z$, the emission release height $H_e$ and ambient wind conditions, the

Gaussian steady-state solution can be used to estimate the concentrations at the point RP ($c_{RP}$) caused by the emitter at RP

while ignoring the effects of urban terrain (Seinfeld and Pandis, 2016). Using the precomputed (scanned) obstacle detection

around RP it can be assessed whether there are obstacles between the emitter and the RP. In case there is an obstacle at a

distance of $d_b$ with height $H_b$ in between the two points, a reduction caused by the obstacle is approximated, as some fraction

of the pollutants is physically unable to disperse to the RP and remains behind the object. The reduction effect should be more

prominent the closer the emission source is to the obstacle. However, as it can be seen from Fig. 3, this distance measure ($d$)

is not readily available (without costly, additional checks) and we approximate it to be $d \approx d_e - d_b$. Using a single

hyperparameter ($a$) we define an approximation of how the distance to the blocker reduces the concentration $c_{RP}$ given by:

| $R_d = 1 - ad,, R_d \geq 0$ | *(1)* |
|---|---|

where the hyperparameter $a$ defines how sensitive the reduction is as a function of distance $d$. Simply put, with a low value of

$d$ the $R_d$ is close to 1 (almost full reduction) and conversely with a maximum distance value ($d = a$) $R_d$ is equal to 0 (no

reduction). The physical interpretation of Eq. 1 is that the emissions originating far behind the barrier can be considered well-

mixed, and the effect of the barrier gradually and continuously loses relevance as a function of the distance.





In addition, we assume that the reduction is proportional to the elevation difference $H = (H_b - H_e), H \geq 0$. The strength of this effect is managed with another hyperparameter ($b$) to estimate the reduction factor due to elevation difference given by:

$$R_H = (H_b - H_e)b, R_H \in [0,1] \qquad (2)$$


Finally, we can approximate the reduced concentration at RP, $c'_{RP}$, with

$$c'_{RP} \approx (1 - R_d R_H)c_{RP} \qquad (3)$$

This simplistic reduction effect modelling (using as few as possible parameters) is applied with the noise barrier located within
the measurement campaign site; the model is also applied to other obstacles such as buildings while acknowledging that the reduction model may not provide accurate results for such more complex objects. Noise barriers are described in OSM data, and it would be technically possible to automatically characterize them as obstacles for the model. Instead, in this study we have manually inserted the noise barrier as a 6.5 m tall construct to the model inputs. In reality, the barrier is thin but due to the way the object detection and the mapping of objects works in the model, the modelled barrier has an artificial width of 4
m (the minimum surface area of any object is 4x4 m).

In this study, we have used values $a$ = 0.0033 m$^{-1}$ and $b$ = 0.1 m$^{-1}$ for the hyperparameters. These hyperparameter values were obtained by using Monte Carlo simulation with the sensor data from NB10m, NB20m and NB40m, focusing on PNC, PM$_{10}$ and LDSA measurements. The hyperparameters that minimized the root mean squared error over all the selected species were chosen.

**2.4.2 Road traffic exhaust emission modelling**

Road traffic emissions are modelled by combining hourly vehicle flow information for individual roads (given by OSM) coupled with vehicle class–specific emission factors. These emission factors are also flow-speed dependent, and the flow speed is also used to introduce instantaneous mixing due to the turbulence caused by the vehicles (Johansson et al., 2022). Individual roads are described as objects in the model with various characteristics. While these objects are not exactly lane-specific, the
different flow directions are most often represented by separate and independent objects. For example, at the campaign site the highway has two separate parallel road objects that characterize the traffic flows to the West and separately to the East (Fig. 4). Both objects characterize the number of lanes, but this information is only used in defining the width of the emission source assuming that the vehicle flows are evenly distributed to the lanes. In this study, the westbound section of the road, which is closer to the measurement campaign site, has a greater impact on the concentrations at the measurement locations.


In the Enfuser model, the average hourly flow counts are described for each road object separately using 24 average flow values for working days (Monday to Friday), another 24 values for Saturdays and finally 24 values for average Sundays. This



characterization with 72 values is defined separately for cars and heavy vehicles. Additional "regional" modifiers (targeting all road objects) are used to address weekly variations (e.g., summer holiday months. In this study, traffic count data from the

nearest DigiTraffic (2024) record were used to refine the average flow information applied in modelling the site area. This localized fine-tuning of the vehicle flows was possible since there is a traffic count sensor very close to the campaign site. As can be seen from the figure, the hourly flows of vehicles to the West and East are clearly different (asymmetric) for cars during working days.

**Figure 4: Hourly average traffic flow information incorporated into the Enfuser model. Up, the overall daily average**
**flows (cars) near the campaign site are shown. Down, the implemented direction-specific hourly traffic flows are shown for cars and heavy vehicles in local time. The first 24 hourly flow counts correspond to working days (Monday to Friday) and the next 48 values correspond to Saturdays and Sundays.**

### 2.4.3 Modelling of non-exhaust emissions from road traffic

The modelling of coarse particle generation (e.g., via the use of studded tyres) and resuspension of the particles is challenging; an overview of the approach has been presented in (Johansson et al., 2022). Recently, the use of machine learning-assisted



modelling of road dust has also been investigated (Kassandros et al., 2023). Without relying on machine learning, however, the model describes a generic weekly pattern for the usage of studded tyres in the modelling area for passenger cars. This information combined with the hourly flows of vehicles (while considering the flow speed) provides an upper limit function for the coarse particle fraction emissions. Further, we assume that heavy vehicles also generate coarse particles through wear and tear. This upper limit function is then limited (scaled down) based on road surface moisture. Simply put, for a road that is wet or covered in ice and snow the upper limit function is reduced to near zero values. As a complication, the Gaussian models struggle to consider dynamic phenomena such as the generation of resuspension particles in the past; based on our previous studies of diurnal variability of $PM_{10}$ concentrations in urban traffic monitoring stations we have learned to incorporate the vehicle flow information from the past two hours to obtain better agreement with modelled and measured concentrations and to make hourly $PM_{10}$ predictions less sensitive to the most recent vehicle flows nearby. Finally, the modelled resuspension component is being adjusted on an hourly basis according to the recent measurement evidence via the data assimilation routine. As described by Johansson et al. (2022) the data assimilation corrections cannot address local biases (e.g., an especially dusty road) but modify emission factors and background concentration across the whole modelling area.

### 2.4.4 LDSA, BC and PNC modelling

In this study, we present results for LDSA, BC and PNC concentrations for the first time in Helsinki with the Enfuser model. The modelling approach for these new species is fairly similar to the modelling of, e.g., $PM_{2.5}$ with a couple of exceptions. First, the emission factors are not well known for LDSA and PNC and we rely on proxies that have been based on $PM_{2.5}$ emissions sources. Secondly, the SILAM CTM model does not provide regional background concentrations for LDSA or PNC and thus, we again use a proxy based on SILAM $PM_{2.5}$ background. As these preliminary emission factors and background estimates are almost certainly biased, we rely on Enfuser's data assimilation method to gradually adjust these to the levels that provide the best fit against the measurement evidence. The stabilization of these parameters takes time, and this is one of the reasons the modelling time span has been set to begin on 1st of January. Finally, particle number concentrations are impacted by meteorological conditions, e.g., affecting coagulation (Gani et al., 2020). Since the measurement input for PNC does not include size distributions the modelling cannot address such effects.

### 3. Results and discussion

### 3.1 Influence of the noise barrier based on measurement results

The influence of the noise barrier as a function of distance was analyzed using the measurement data, for which the results are shown in this section. The gradients (i.e., concentration changes with distance) of the main pollutants $PM_{10}$ (> 0.6µm), $PM_{2.5}$(> 0.6µm), PNC, LDSA, BC, and $NO_2$ are shown in Fig. 5. All measured pollutants show a clear decreasing gradient in the open area, with the most significant decreases observed between O10m and O20m poles. The steepest gradients were observed for particulate-related pollutants $PM_{10}$ and $PM_{2.5}$ which is in line with previous findings in the literature. For example, Zheng et



al., (2022) found that larger particles have steeper and more pronounced roadside-decreasing gradients compared to smaller
particles. The $PM_{10}$ and $PM_{2.5}$ gradients were likely to have been affected by the strong street dust season that enhanced the coarse particle concentrations on the highway compared to the background concentration. Additionally, the $PM_{2.5}$ gradient might have been slightly overestimated as the sensors measured only particle sizes > 0.6 μm and therefore the street dust contribution to $PM_{2.5}$ concentrations was exaggerated. The gradients of the large particles might also be due to the larger particles having slower dispersion, more efficient deposition and shorter removal times from the atmosphere compared to the gaseous pollutants and smaller particles. The effect of noise barriers on $NO_2$ concentrations was less pronounced and the effect of the noise barrier was diminished already at 20 m from the highway. Additionally, the decrease for PNC was notable despite measurements being available only at O20m and O40m poles (measurements from O10m were not available and likely would have shown the highest concentrations due to the proximity to the highway).

Behind the noise barrier, all the pollutant concentrations were lower when compared to the open area with no clear decreasing gradients for the pollutants. Slight decreasing gradients even behind the noise barrier were detected for PNC, LDSA and $NO_2$, but For $PM_{10}$ and $PM_{2.5}$ there were no gradients behind the noise barrier and the lowest concentrations were seen straight behind the noise barrier.






**Figure 5: A) PM₁₀, B) PM₂.₅, C) PNC, D) LDSA, E) BC, and F) NO₂ gradients in the open area and behind the noise barrier. In panel C the PNC gradient has been measured with AQ urban sensors and PNCs measured with CPCs have been presented with 'X'. IN panel E the concentrations at 20 m have been measured with AE51 and concentrations at 10 and 40 m with Observair sensors. The gradients have been calculated over all the measured data. The data coverages for each parameter are presented in supplement S1.**

**3.1.1 Characteristics of aerosol at the highway**

During the measurement period, a mobile laboratory was used to measure pollutant concentrations on the highway for two days: March 20th between 12:45 and 14:50 and March 24th between 11:20 and 15:15 (local time). The corresponding stationary data for these hours was extracted and compared with the on-road measurements. Figure 6 represents the concentration gradients in the open area for BC, LDSA, NO₂ and PNC. BC data from all the poles were available during both measurement

days. For both days, BC concentrations were significantly higher on the highway compared to the roadside concentrations. On the roadside, the BC concentrations have a similar gradient to the one seen in Fig. 5, with concentrations decreasing steeply



from O10m to O20m but either slightly decreasing (or even increasing) from O20m to O40m. For LDSA, data from the O20m pole on March 24th was missing. However, significantly higher LDSA concentrations were observed on the highway compared to the roadside on both days. The gradients of LDSA and BC follow a logarithmic curve, consistent with earlier roadside observations by (Enroth et al., 2016; Zheng et al., 2022). Coincident stationary measurements of $NO_2$ were only available for March 20th. On this day, $NO_2$ concentration was only slightly higher on the highway compared to the O10m pole, with a steeper decrease between O10m and O20m. This trend may partly reflect the conversion of traffic-emitted NO to $NO_2$, leaving a higher proportion of $NO_x$ as NO on the highway. In the case of Fig. 5, it was speculated that the decrease in PNC could be steeper closer to the highway if measurements were available. In Fig. 6 this seems to be the case as on the 20th of March the decrease of PNC between the highway and O20m is greater compared to the decrease between O20m and O40m poles. On the 24th of March, the PNC data from the O20m pole was not available, however, the concentrations on the highway and at O40m were similar to those measured on the 20th of March.

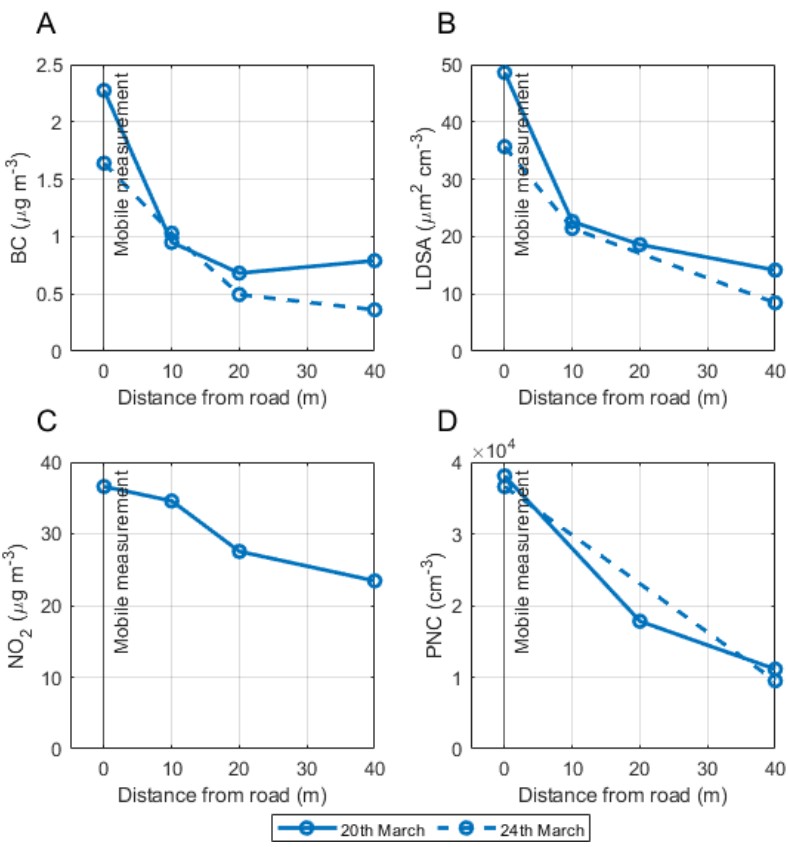

**Figure 6: A) BC Gradient with mobile measurement. B) LDSA gradient with mobile measurement. C) NO₂ gradient with mobile measurement. D) PMC gradient with mobile measurement. The mobile measurement results have been added to the figure as 0 m points representing the side of the road.**



### 3.2 Drone measurements

Vertical differences in PNC at the site were studied with drone measurements. These measurements were performed at the same time as the mobile measurement. The PNC was measured using a multicopter, flying along the highway from behind the
noise barrier to the open area at two different heights: 2 m and 15 m. In Figure 7 the results from these measurements are presented separately in two panels for the two measurement days 20th and 24th of March. Each day includes four boxplots, corresponding to the heights of 2 m and 15 m, and with separate plots for the flight paths behind the noise barrier and in the open area. The measurement data nearer than 10 m from the edge of the noise barrier was left out of the analysis both in the open area and behind the noise barrier.

Figure 7 shows that the median concentrations, indicated by the red horizontal line in the figure, were slightly lower at a height of 15 m compared to 2 m on both days. The effect of the noise barrier on the concentrations measured with the drone seemed to be smaller compared to the measured gradient presented in Fig. 5. Also, the lowest concentrations were measured in an open area at an elevation of 15m. Notable was also the large variability of the PNC with the measured values varying between a couple of thousands to more than 50 000 cm$^{-3}$ during both days, although the measurements only consisted of a total of 8
flights, 4 on each day. Each of the flights lasted for 5 to 10 minutes and during both days. The pauses between flights were between 5 to 20 minutes.

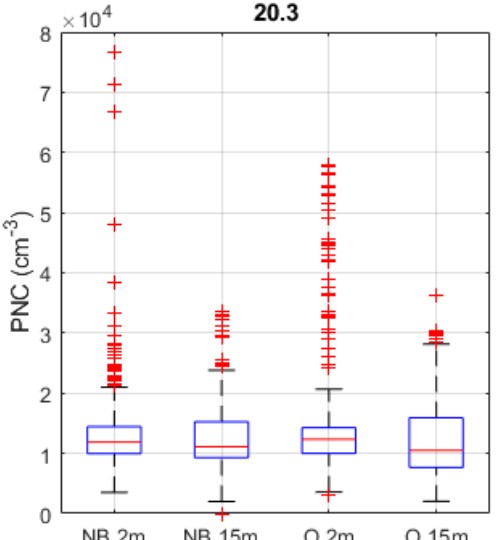
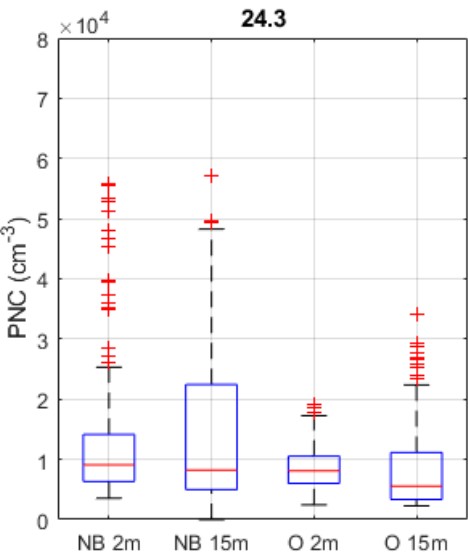

**Figure 7: Boxplots of PNC measured with a drone in the open area (O 2m, O 15m) and behind the noise barrier (NB 2m, NB 15m)**
**separately for the 20th and 24th of March. The median values are indicated with horizontal red lines, the blue box resembles the lower and upper 25th and 75th percentiles, the whiskers represent the lowest and highest values not considered outliers and the outliers are marked with red plus marks.**




## 3.3 Analysis of measured and modelled concentrations

In the next sections, we present various results where modelled and measured pollutant concentrations have been compared.
The modelled and observed average pollutant concentrations for BC, $PM_{10}$, LDSA and PNC have been shown in Fig. 8 for the
6 or 4 measurement poles depending on the pollutant species over the whole measurement campaign period. The split between
emission source categories has been shown for the model predictions in the form of staggered columns. With regards to the
modelled emission source categories "RWC" stands for residential wood combustion, and the category "Other" stands for a
collection of minor source categories such as shipping, power plants and aviation. It should be noted that the averaging period
varies between the measured pollutant species and as such the cross-comparison of e.g., BC and $PM_{10}$ at pole O10m should be
avoided. The model predictions are affected by the measurement input in the whole Helsinki metropolitan area via the data
assimilation procedure while the campaign measurements have been excluded from the data assimilation. However, the
campaign measurements for $PM_{10}$, LDSA and PNC behind the noise barrier were utilized in a separate prior study to optimize
the necessary hyperparameters for Eqs 1-2. This means that the hourly emissions factors and regional scale background are
constantly being adjusted based on the measurement evidence that is obtained outside of the campaign site. The model
evaluation against the reference air quality stations that provide this input (up to 12 stations depending on pollutant species)
has been excluded in this paper and will be published separately.

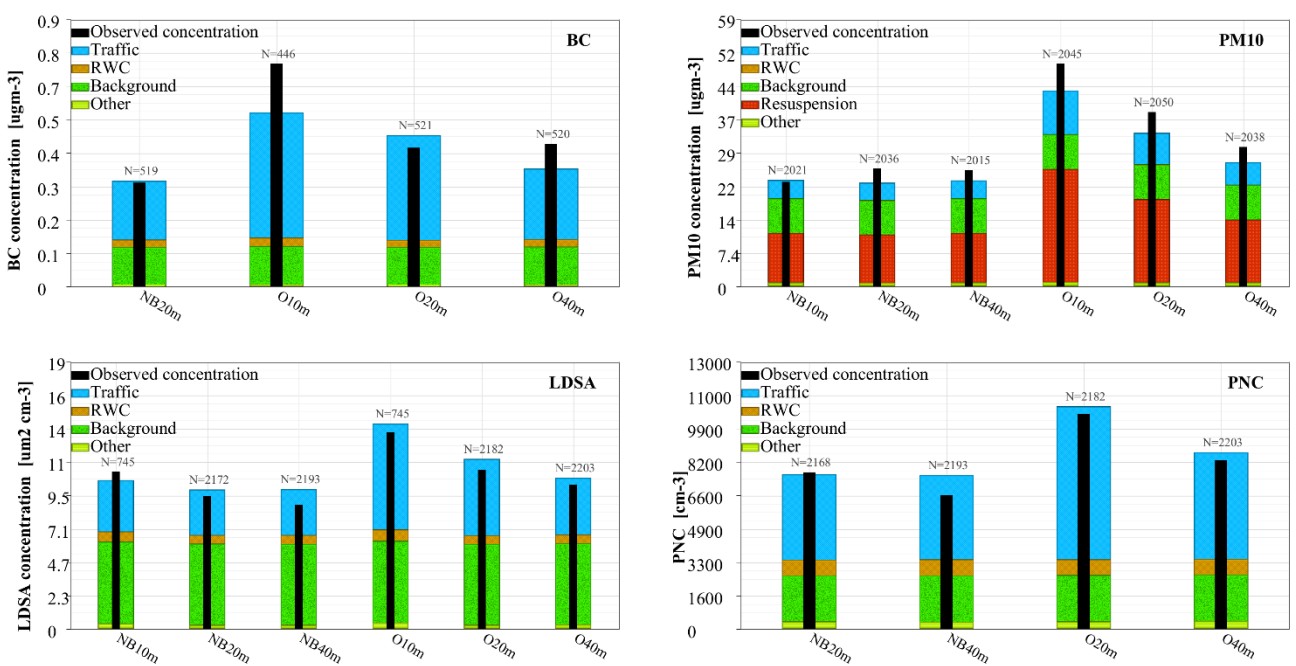

**Figure 8: Modelled and observed average BC, $PM_{10}$, LDSA and PNC during the campaign at the different measurement poles. The**
**number of hourly data points forming the average for each pole and pollutant species has been shown as a number on top of the**
**column.**



The most notable difference between the modelled concentrations and the measurements is that the model underestimates $PM_{10}$ concentration in all but one of the 6 measurement locations (NB10m). At NB10m the modelled concentration is almost one-to-one with measured concentrations. The largest deviation of the model with measured concentrations in the case of BC is

observed at O10m, where the average BC concentration is significantly underestimated. Also, the model seems to expect a constant decrease for BC in the open area with increasing distance from the highway, whereas the measured concentrations were slightly higher at O40m compared to O20m. For $PM_{10}$ the modelled and measured concentrations seem to agree mostly well with no notable gradients behind the noise barrier and similar gradients in the open area with each other. Only the observed concentrations especially in open area were higher. In the case of LDSA and PNC, the trends and concentrations of measured

and modelled concentrations were very similar. Notable is that the traffic-related particles had quite significant contributions to all BC, LDSA, PNC and $PM_{10}$. Additionally, in the case of $PM_{10,}$ the effect of traffic is accompanied by resuspended particles that corresponded to half of $PM_{10}$ observed behind the noise barrier and more than half of the $PM_{10}$ in the open area. The contribution of resuspension was not visible in the case of BC, LDSA or PNC. In the case of BC and PNC, the contribution from traffic dominated the contribution from the background but in the case of LDSA the contribution from the background

exceeded the contribution of traffic at all other poles but the O10m pole.

## 3.4 Measured and modelled daily mean concentrations

The measured and modelled pollutant concentrations for the main measured pollutants BC, $PM_{10}$, LDSA, PNC, and $NO_2$ have been compared in Fig. 9 in the form of daily averages. On the left-hand side, we show results for the pole behind the noise barrier that is closest to the highway and has available data for the pollutant. On the right-hand side, we show results for the

pole nearest to the highway in the open area that had data available for the current pollutant. For brevity, the other remaining measurement locations are omitted as they resemble one of these two presented plots. We will use this style of representation and selection of plots in other figures as well. With regard to the presented results in this work, we focus on the less studied BC, PNC, LDSA and the more commonly studied $PM_{10}$.

As it can be seen from Fig. 9, the amount of BC data is relatively low and thus the results attained for BC should be considered

slightly more uncertain compared to results related to the other measured variables. For BC the daily mean concentrations varied between approximately 0.1 and 0.8 µg m$^{-3}$ behind the barrier and between 0.2 and 1.8 µg m$^{-3}$ in the open area. During the measurement period, frequent increases were seen in the $PM_{10}$ concentrations, especially during the dry periods, in March and April. These episodes could be affected resuspension of road pavement ground by studded tyres and salt that are used on the roads in wintertime to prevent slippery road conditions. Which has been shown to result in frequent road dust episodes in

springtime next to roads in Finland (Pirjola et al., 2010). The frequency of $PM_{10}$ episodes decreased in May, likely due to effective road cleaning in the area, and natural cleaning of the roads by rain and wind, with the reduction in the use of studded tyres. During the measurement period, daily averages for $PM_{10}$ reached 180 µg m$^{-3}$ in the open area. But starting from late April these peaks were significantly lower with $PM_{10}$ staying below 70 µg m$^{-3}$. The high concentrations in early April may have been enhanced due to the lack of rain. Behind the barrier, the concentrations were generally lower with concentrations





reaching over 90 µg m$^{-3}$ in early April but later staying mostly below 40 µg m$^{-3}$. For LDSA the observed concentrations behind the barrier and in the open area were at similar levels, being only slightly lower behind the noise barrier with a daily average reaching up to 23 µm$^{-2}$c m$^{-3}$. For PNC the concentrations were somewhat smaller behind the noise barrier, especially the observed daily maximum concentrations that reached 21000 cm$^{-3}$ behind the noise barrier compared to 30000 cm$^{-3}$ in the open area.




**Figure 9: Modelled and observed daily average pollutant concentrations of BC, PM10, LDSA and PNC for selected measurement locations (NB20m or NB10m and O20m or O10m).**





Fig. 9 shows that the model has the tendency to underpredict PM$_{10}$, LDSA and PNC before the end of March. After April,

however, the model has the tendency to overpredict concentrations, especially with PNC and LDSA. The underestimations of concentrations were most significant during days with the highest concentrations in March. This phenomenon was visible in all the pollutants but most clearly in PM$_{10}$. These further underline that there seem to be some difficulties in modelling the road PM$_{10}$ emissions during snow cover and the use of studded tyres. During the low-concentration days, the agreement between the modelled and measured results was better. Interestingly, the weather also got significantly drier approximately at

the same time (Fig. 2). For BC the change of model accuracy could not be evaluated as the data is limited to only March. However, during that time, the time series of the observed and predicted concentrations were very similar behind the noise barrier. However, in the open area, the model seems to underestimate the BC concentrations. When comparing the modelled and measured daily concentration averages it is seen that the model agrees with the measured data better during April and May compared to March.

In Table 1, two chosen accuracy indicators (Factor-of-two, Pearson correlation) for the hourly modelled concentration against the sensor measurements (all locations) have been presented. There was a strong hourly variability in the measurement data that could not be captured by the model. None of the 6 measurement locations stands out, showing that the modelling accuracy was evenly matched in all locations. The most challenging hourly variability to model could be observed with O10m measuring PM$_{10}$; approximately half of the time the ratio of the measurement and the observed concentration was between 0.5 and 2.

LDSA had clearly the highest agreement in terms of Pearson correlation as well as with the Factor-of-two indicator, while the worst correlation could be observed with PNC. Interestingly, the modelling of both LDSA and PNC relies on proxy information for emission factors and data assimilation-based learning, and this approach works well with LDSA but is less effective with PNC.

**Table 1: Accuracy indicators for hourly modelled and measured concentrations for PM$_{10}$, LDSA, PNC and BC. FAC2 stands for Factor-of-two, and PCC stands for Pearson correlation coefficient. The definitions of these indicators can be found in the Supplements.**

|  | PM$_{10}$ | | LDSA | | PNC | | BC | |
|---|---|---|---|---|---|---|---|---|
|  | FAC2 | PCC | FAC2 | PCC | FAC2 | PCC | FAC2 | PCC |
| NB10m | 0.60 | 0.53 | 0.88 | 0.56 |  |  |  |  |
| NB20m | 0.61 | 0.55 | 0.89 | 0.62 | 0.73 | 0.40 | 0.68 | 0.53 |
| NB40m | 0.63 | 0.57 | 0.88 | 0.64 | 0.73 | 0.38 |  |  |
| O10m | 0.51 | 0.55 | 0.87 | 0.67 |  |  | 0.61 | 0.55 |
| O20m | 0.56 | 0.60 | 0.86 | 0.63 | 0.71 | 0.50 | 0.65 | 0.55 |
| O40m | 0.62 | 0.62 | 0.88 | 0.64 | 0.73 | 0.43 | 0.64 | 0.49 |





**3.5 Diurnal variability of pollutant concentrations**

The average diurnal profiles for the selected pollutants are presented in Fig. 10. Similar plots for $PM_{2.5}$ and $NO_2$ are included in the supplementary material. Additionally, the supplementary material provides a breakdown of the modelled diurnal profiles by emission source categories. Diurnal profiles for the measured and modelled results are presented separately for BC, $PM_{10}$, LDSA and PNC for all available poles. It is important to note that these pollutants do not have completely overlapping time series, which might contribute to slight differences in their diurnal patterns. This issue is particularly relevant for BC, which

has data only for March. The modelled contributions from different sources to the diurnal profiles are presented in supplement S1 and discussed in the related text.





**Figure 10: Average diurnal variation of modelled and measured concentrations for BC, PM$_{10}$ PNC and LDSA presented separately for all available measurement poles.**

When compared to the measured and modelled BC diurnal profiles, the modelled concentrations were consistently lower at all the poles and had similar diurnals with traffic having minor peaks at the morning and evening rush hours at around 8 am and 4 pm. The observed BC concentrations peaked at approximately 1.5 µg m$^{-3}$ at 4 pm and reached a minimum of around 0.2 µg m$^{-3}$ at 3 am. In contrast, the modelled BC concentration had a maximum of 1.0 µg m$^{-3}$, notably lower than the observed values. Additionally, the observed BC concentrations showed a second peak around 9 pm, with a concentration of 1.3 µg m$^{-3}$, which was not observed in the modelled BC diurnal. Both modelled and measured BC concentrations showed a similar minimum of





0.2 µg m$^{-3}$ during the early morning hours, with negligible differences between poles at this time. A significant difference between the modelled and measured concentrations lies in their gradients. In this sense, for the modelled concentrations, the decrease was gradual with increasing distance from the highway, and concentrations were lower behind the noise barrier. In
contrast, for the observed concentrations, most of the reduction occurred between the O10m and O20m poles, with relatively similar concentrations at the other poles.

For PM$_{10}$ the modelled and measured diurnals had similar shapes with concentrations reaching a minimum in the early morning hours and being elevated with the traffic. The measured PM$_{10}$ reached its maximum of around 85 µg m$^{-3}$ during the afternoon
rush hour and the minimum value below 20 µg m$^{-3}$ during the early morning hours. The modelled PM concentrations reached their maximum of approximately 75 µg m$^{-3}$ at 10 am, and minimum value below 20 µg m$^{-3}$ similarly to measured PM$_{10}$ in the early morning hours. The diurnal pattern of all the poles was very similar with only the concentrations decreasing with the increasing distance from the highway.

The observed LDSA concentration had a bimodal diurnal with two distinct modes at 8 am and 2 pm with concentrations of 21 and 22 µm$^{-2}$ cm$^{-3}$, respectively. Whereas the modelled LDSA diurnal had only one clear mode at 7 pm with concentrations of 25 µm$^{-2}$ cm$^{-3}$. During the afternoon the modelled LDSA was also elevated with concentrations around 18-20 µm$^{-2}$ cm$^{-3}$, but no clear mode was visible. Both the observed and modelled LDSA concentrations reached the minimum at around 4-5 am with the concentrations around 9 µm$^2$ cm$^{-3}$ for both.


The peak for observed diurnal PNC was around 19000 cm$^{-3}$ and it was observed around 8 am. The second mode peak of the PNC was observed at around 15000 cm$^{-3}$ at 2 pm. The modelled PNC showed similar diurnal and maximum values to the measured ones, with a maximum of around 20000 cm$^{-3}$ at 7 am and an afternoon peak of 17000 cm$^{-3}$ at 4 pm. Both the modelled and observed diurnals had their minimum in the early morning hours and PNC of around 5000 cm$^{-3}$. Notably, in the case of
observations NB40 had a lower concentration than NB20 during the whole day, whereas in the case of modelled PNC, the NB40 and NB20 had very similar concentrations throughout the day. In the case of observed concentrations, there was a reduction in PNC when moving further away from the highway and the lowest PNC were measured behind the noise barrier.

Overall, the modelled and measured diurnal profiles were quite similar, and the largest differences were observed for the BC
concentrations. For all the pollutants the lowest concentrations were measured during the early morning hours before the traffic started to increase and the highest concentrations were during the daytime indicating strong contributions from traffic. All the measured pollutants BC, PM$_{10}$, LDSA and PNC also reached similar concentrations at each of the poles during the early morning hours. This enhances the trustworthiness of the sensor results, as during this period when traffic's contribution to concentrations was minimal, the sensors were effectively performing collocated measurements. Under these conditions, the
sensors should display similar concentration readings if the instruments are functioning properly. Additionally, the model



results during this time were on a very similar level to the measured results which implies that both the model and sensors were effective in measuring the background concentrations.

### 3.6 Modelled and measured concentrations as a function of wind direction

In Fig. 11 conditional averages as a function of wind direction for measured and modelled concentration are shown for $PM_{10}$, LDSA, PNC and BC. Again, on the left-hand side, we show the concentrations behind the noise barrier and at the open area on the right-hand side. HARMONIE NWP meteorology was used in the modelling and thus the presented averages are based on HARMONIE wind directions. There are an unequal number of data points for different wind directions; at worst for BC measurements there are only 20 data points available to compare measured and modelled hourly concentrations when the wind direction is between 330 to 360° degrees. The measurement site was located on the northern side of the highway that was oriented in the West-southwest to East-northeast direction and therefore when the wind is blowing between approximately directions of 90 to 225° the air mass passes over the highway.

As expected, the traffic emissions (and resuspension of particles) were most prevalent with wind directions between 90 to 270° that corresponded to the location of the highway from the measurement site. As can be seen from the results for NB10m the model overestimated the blocking effect of the barrier (i.e., the model underestimated concentration) when the wind direction was orthogonal to the highway. Similarly, when the wind was blowing directly from the highway the $PM_{10}$ concentrations were underestimated at O10m. When the wind was not blowing from the direction of the highway, the model under predicted $PM_{10}$, at the O10m pole. This could indicate that the model underestimated the effect of speed on $PM_{10}$ emissions from the highway, as the high-speed limit at the site is 100 km/h which is unique in the Helsinki metropolitan area measurement locations and no similar underestimations are observed at reference stations.

The model also underpredicted the $PM_{10}$ concentrations between 300 and 30 degrees in the open area. Similar underprediction was not observed behind the noise barrier. The under-prediction of concentrations in the wind directions 300 to 360 was not limited to $PM_{10}$ but was also observed for LDSA, PNC and BC. For LDSA PNC and BC underestimation of concentrations was observed also behind the noise barrier. The most notable underestimation was observed for BC at O10m with wind directions between 300 to 360 degrees (which was the direction of the storage building behind the measurement area), possibly indicating that there was a burning source in this direction that was not considered by the model. In the case of BC, the source could have been car workshops on the roadside of the storage building situated behind the measurement site in this direction. The cars running in front of these workshops could have caused some elevated concentration when the wind was blowing from this direction.

Underestimation of concentrations at O10 might also indicate contributions from traffic to also when the wind was not coming from the direction of the highway that the model was not able to capture. Also, any dry surface with resuspension particles can act as a dust source given sufficient meteorological conditions but this possibility has been omitted from the model. Despite



the observed differences, the simplistic treatment of noise barrier in a Gaussian dispersion model still results in a reasonable agreement between measured and modelled concentrations.

**Figure 11: Modelled contributions of different sources to PM$_{10}$, LDSA and PNC and observed concentrations at different poles as a function of ambient wind direction (HARMONIE NWP). The first wind direction (< 30°) contains data points when the wind has blown from a direction that is between North (0°) and 30 degrees from North towards East.**

### 3.7 NO$_2$ Passive samplers

To capture the effect of exhaust gasses around the measurement area. The NO$_2$ concentrations were measured via the passive samplers (locations shown in Fig. 1). In Fig. 12 the measured and modelled NO$_2$ concentrations are presented for data averaged




over February, April and May. The results during March suffered from quality issues and have therefore been omitted from this paper. The highest observed and modelled concentrations are seen for PAS_6m which was located on the roadside of the noise barrier to the proximity of the highway. This location is especially challenging for Gaussian models, as the close by noise

barrier can affect the micrometeorology near the road. Overall, the modelled and measured $NO_2$ concentrations were in good agreement, especially in the locations behind the noise barrier. The largest differences were observed for the poles closest to the highway (PAS_010m, PAS_O20m and PAS_6m). The measured $NO_2$ averages were consistently lower than the measured concentrations. Further, in open areas, the decreasing trend of $NO_2$ is quite insignificant while the model suggests a more intuitive and expected decreasing trend as a function of distance. Additionally, the $NO_2$ concentrations measured at the poles

with passive samplers were lower compared to the measurements used for calculating the gradients. The lowest concentrations both modelled and measured were seen for PAS_100m which was the pole situated furthest from the road.

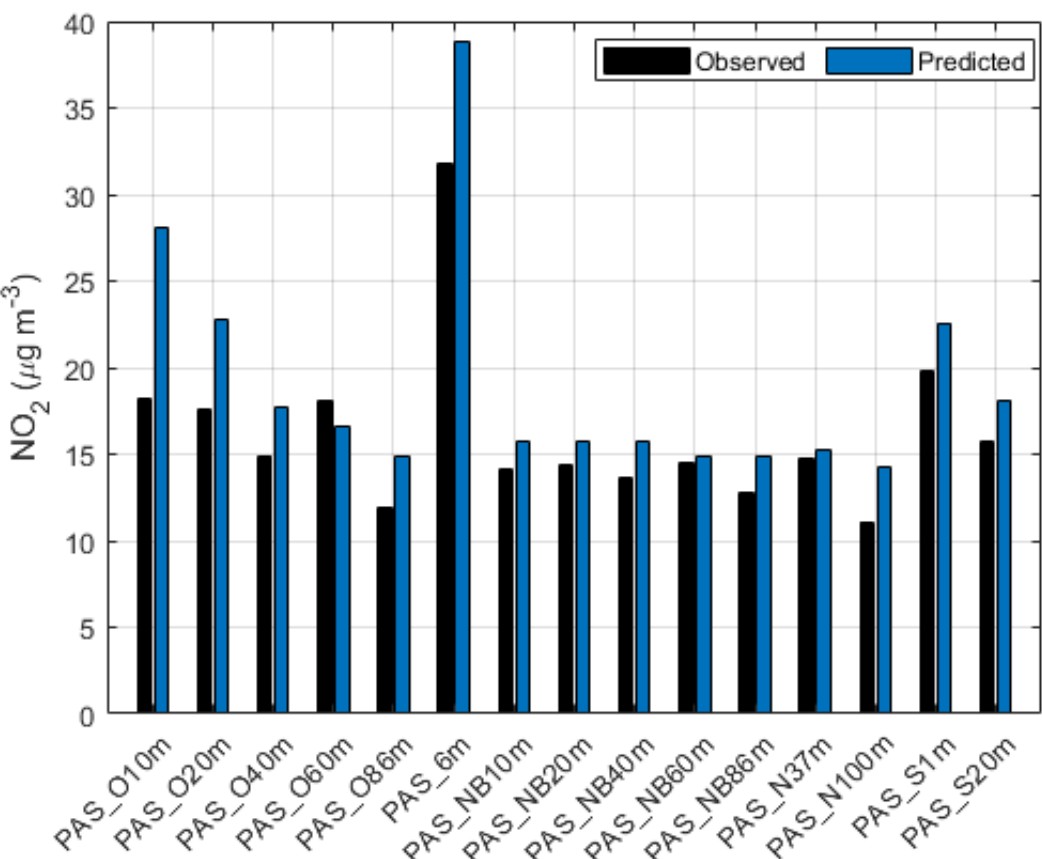

**Figure 12: Observed concentrations (μm m$^{-3}$) with passive $NO_2$ samplers compared against Enfuser predictions for data averaged over, February, April and May.**



**3.8 Modelled geographical distribution of pollutants**

In Fig. 13 the modelled concentrations (at 2 m above ground) are presented around the Helsinki metropolitan area and in more detail around the measurement site. We have selected to show $PM_{10}$ monthly average concentrations for April. However, similar geographical distributions are available for all focused pollutant species and for all months during the campaign period. The presented monthly average has been processed from hourly concentration distributions as the Enfuser model estimates

these as its main, publicly available output.

The model predicted the highest concentrations overall in the Helsinki area around the largest highways with the concentration scale reaching up to 100 µg m$^{-3}$. Overall, the concentrations were seen to decrease quite sharply with increasing distance from highways on a large scale. In the zoomed-in figure of the measurement location, the noise barriers are seen in the figure as

light green lines (very low concentrations at the location of the barrier) on the northern side of the highway. In the figure the effects of utilizing Eqs. 1-3 can be seen in the area where the gap in the noise barrier is located; the modelled concentrations in the gap are clearly higher than the ones behind the barrier. Further, the modelled concentration behind some of the buildings is also reduced due to the obstacle reduction method, showing that the capability is not limited to specific obstacles such as noise barriers.




**Figure 13: Modelled average PM$_{10}$ concentrations during April 2023 in the overall Helsinki metropolitan area and near the measurement campaign site.**

## 4. Conclusions

Overall, the noise barrier of 6.5 m in height was found to be effective in reducing the measured concentration of pollutants behind the barrier. The concentrations behind the noise barrier were lower for all the measured pollutants PM$_{10}$, PM$_{2.5}$, PNC, LDSA, BC and NO$_2$. The difference between the open area and the area behind the noise barrier was the largest for PM$_{10}$ and



the smallest for $NO_2$. Measurement data also showed that concentrations of all pollutants decreased as a function of distance from the highway with the steepest gradients being observed nearest to the highway. The decreasing gradient was strongest

for the $PM_{10}$ and least significant for NO2 behind the noise barrier the decreasing gradients were less clear and for example, $PM_{10}$ had the lowest concentrations closest to the road at NB10 pole with higher concentrations at NB20 and NB40. Indicating that the noise barrier effectively blocks the dispersion of $PM_{10}$ from the road but when the distance to the road increases the concentrations from the open area get mixed with the airmass again elevating the concentrations.

Modelled concentrations of all pollutants showed good agreement with measurements. This success can partly be attributed to tailored modelling of nearby traffic flows, which were adjusted to match the real vehicular flow data. This flow customization was performed separately for the two traffic flow directions and for passenger cars and heavy vehicles. The largest difference between modelled and measured pollutants was observed for $PM_{10}$ which was underestimated compared to the measurement near the campaign site. This may indicate that the model underestimates the speed dependency of coarse particle emissions, as

the high-speed limit of 100 km/h at this site is unique in the Helsinki metropolitan area and no similar underestimation was observed at other reference stations (not shown in this paper). Another notable underestimation by the model was observed with BC in the pole closest to the highway in the open area.

The noise barrier was considered in the modelling by defining it as an obstacle. A statistical reduction for concentrations was

applied based on the distance from the emitter to the obstacle and the height difference between the obstacle and the emitter. The simplified modelling approach captured the real-life effect of the noise barrier; however, more research would be needed to generalize and properly parametrize the approach. The simplistic reduction model used two hyperparameters that were calibrated based on the measurement data, using Monte Carlo simulation. In this simulation, the parameter values were varied to find an optimal state that results in as low as possible prediction error in terms of RMSE for PNC, $PM_{10}$ and LDSA. Only a

handful of values for each hyperparameter were tested, as the overfitting of a simple statistical reduction model with only a few measurement locations providing calibration data should be avoided. Another limitation of the optimal parameter assessment comes from the fact that the barrier height was constant, i.e., the applicability of the simple model remains untested with different barrier heights. Nevertheless, the simplistic reduction model has room for improvement provided that a more thorough training set is available. One option is to utilize the CFD models' output and calibrate a more generalized reduction

method if steady-state concentrations provided by the CFD model (e.g., a LES model) are realistic proxies for true concentrations.

The modelled $NO_2$ average was compared against passive $NO_2$ measurements. According to the results these were in general agreement with the exceptions in two sensors close to the highway (PAS_6m and PAS_O10m). With these two sensors, a

strong overprediction was seen during February and April, but not during May. These discrepancies may have been the result of simplified in-plume $NO_x$-Ozone photochemistry that is being used in Enfuser and meteorological effects affecting the

passive sensors. It is also possible that the passive sensor measurements during April were biased indicators for true concentrations. In this paper, we have demonstrated the benefits of combining measurements and modelling approaches in the analysis of air quality. The measurement data can be used to improve modelling capabilities, and the modelling results help to

understand and analyse obtained measurement data.

**Author contributions**

SH, LJ, JVN, TP, and HT contributed to conceptualization. SH, LJ, JVN, VS, VL and JACV handled data curation. SH and LJ performed the formal analysis of the data. TP and HT were responsible of the funding acquisition. SH, JVN, VS, JACV, KL, VL, KD and DB handled the investigation. LJ contributed to the methodology, model development and used the model

for output creation. TR, HEM, TP and HT were responsible for project administration. SH and LJ handled the visualization and writing the original draft. All of the authors also contributed to review and editing review and editing.

**Acknowledgements**

We thank Jussi Hoivala and Jyrki Widenius for their help in experiments. This work was supported by Technology Industries of Finland Centennial Foundation (Urban Air Quality 2.0 project), European Union Horizon 2020 research and innovation

programme under grant agreement No 101036245 (RI-URBANS), European Union's Horizon Europe research and innovation programme grant agreement No 101096133 (PAREMPI), Academy of Finland via the project Black and Brown Carbon in the Atmosphere and the Cryosphere (BBrCAC) (decision nr. 341271) and Flagship Funding (grant no. 337552, 337551, 337549).

**Code availability**

Source code of the Enfuser model is publicly available via GitHub under the MIT license: https

://github.com/johanssl/EnfuserMIT.git. The repository contains the necessary code and input data for operative modelling of air quality in the Helsinki metropolitan area.

**Data availability**

Data produced for the Helsinki Metropolitan Area by the model is publicly available via the Open Data portal of FMI (FMI, 2024).



**Supplement**

**Competing interests**

At least one of the (co-)authors is a member of the editorial board of Atmospheric Chemistry and Physics.

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
