# Peer review of "Measured and modelled air quality related effects of a noise barrier near a busy highway"

_EGUsphere, 2025_

## Author Comment (AC1)

We thank the reviewers for taking the time to read the article and for their valuable comments on our paper. To facilitate the revision process, we have copied the reviewer comments (in black text), and our responses are in blue font. We have responded to all the reviewer comments and made alterations to our paper.

**Reviewer 1**

The manuscript describes a thorough study on the effect of a 6.5 m high noise barrier at a major highway on air pollutant levels behind the barrier. In my initial assessment, I raised multiple concerns that have been adequately addressed by the authors. The manuscript has improved significantly and I only have a few minor comments on the current version:

Line 46: brackets are missing for the citation of (WHO, 2021)

AR: The brackets are now added:

**However, ultrafine particle number concentration (PNC) and elemental carbon (EC) are also included in good practice statements and systematic measurements of them are encouraged (WHO, 2021).**

Line 55: do you mean "…at 5 m behind…"?

AR: Yes, this is now corrected in the text:

**In an earlier study, noise barriers have been found to reduce $NO_x$ concentrations by 23 % at 5 m behind the noise barrier (Tezel-Oguz et al., 2023).**

Lines 101-103: For how long did the passive samplers collect before being evaluated?

AR: The samplers were measuring for 4 months between February and May, and the results were measured monthly. The data from the march was excluded for data quality reasons. This is now stated in the text:

**The passive samplers were measuring from February to May. The results were calculated monthly, but the March data were excluded because of data quality issues.**

Line 106: The AE51 is from microAeth, not TSI

AR: Yes, this is a mistake. AE51 is an instrument from Aethlabs, not TSI. This is now corrected in the text:

**Following air quality parameters related to street dust and exhaust gases were measured: $PM_{10}$ and $PM_{2.5}$ (particle mass with diameter < 10 or 2.5μm, Vaisala model AQT530, Note: AQT530 Vaisala measured particle size > 0.6 μm, Petäjä et al. (2021)), $NO_2$ (AQT530, Vaisala; IVL type passive samplers), black carbon (BC; AE51, AethLabs; ObservAir, DSTech), particle number concentration (PNC, AQ Urban, Pegasor) and lung deposited surface area (LDSA; AQ Urban, Pegasor; Partector, Naneos).**

Line 110: According to the manufacturer's website, the A30 is EN-16976 compliant, i.e. it should have a lower cut size at 10 nm, not 7 nm. Have the temperatures been adjusted to shift the cut size to 7 nm?

AR: Thanks to the reviewer for pointing this out. The reviewer is correct that the A30 usually has a 10 nm cut size; however, it was adjusted by the manufacturer, upon request, so that both instruments (Brechtel and Airmodus) had the same cut size. We have added a comment to clarify that the instrument was adjusted to that cut size:

**Two CPCs, Airmodus A30 (modified by the manufacturer to the 7 nm cut size; Airmodus Ltd., Helsinki, Finland – briefly described in Wlasits et al., 2024) and Brechtel 1720 (Brechtel Manufacturing, 258 Hayward, USA – described in BMI, 2021) were used to measure total particle number concentrations for particles larger than 7 nm at poles O20m and NB20m, respectively.**

Also, the related references are added to the reference list:

**Brechtel Manufacturing Inc. BMI Model 1720 MCPC Manual (Version 2.2), https://www.brechtel.com/wp625 content/uploads/2021/08/bmi_model_1720_mcpc_manual_v2.2.pdf (2021).**

**Wlasits, P. J., Enroth, J., Vanhanen, J., Pajunoja, A., Grothe, H., Winkler, P. M., & Stolzenburg, D.: Reduced particle composition dependence in condensation particle counters. Aerosol Research, 2(1), 199–206. https://doi.org/10.5194/ar-2-199-2024, 2024.**

Line 117: Was the factor 0.66 consistent for all partector devices and more or less constant over time? Do you have an explanation for the rather large mismatch? Could it be due to different calibrations, e.g. the use of different lung deposition efficiency curves?

AR: Yes, the factor 0.66 was found to be remarkably constant over the measurement period, and it was determined in colocation measurements at 20m poles during the whole measurement period. The authors can not pinpoint one exact reason for the difference in the LDSA concentrations, but one reason could be that the manufacturers may use different calibrations, as the reviewer suggested.

Line 135: C shows the precipitation, and D shows the water layer height.

AR: The caption to Figure 2 is now corrected:

**Figure 21: The weather data including A) Temperature (T), B) Boundary layer height (BLH), C) Rain, D) Water layer height on road (WLH), E) Wind speed (WS) and F) Wind direction (WD) during the measurement period (March 1st to May 31st, 2023).**

Line 203: "e" is not a variable and should thus not be in italic.

AR: This is now corrected:

**Let us consider an emission source (e) at some location near the RP for which the emission release rate [$\mu gs^{-1}$] for a pollutant species is known.**

Line 356: I am not a native speaker, but "not considered outliers" sounds a bit awkward in this sentence. I'd suggest rephrasing to "…lowest and highest values considered for data evaluation…"

AR: The sentence is now rephrased as suggested:

**The median values are indicated with horizontal red lines, the blue box resembles the lower and upper 25th and 75th percentiles, the whiskers represent the lowest and highest values considered for data evaluation, and the outliers are marked with red plus marks.**

Line 422-423: Couldn't it also be that the model underestimates the cleaning effect of precipitation? Especially April, but also May were significantly drier than March.

AR: By looking at our modelling results vs. station measurements in the Helsinki region (i.e., not the campaign measurements; these results have not been presented in the paper), we see no clear evidence of this happening. Presumably, this cleaning effect would be most evident for $PM_{10}$, but we do not see clear overpredictions during notable precipitation. As we have notable underpredictions during March (which had higher precipitation than April and May), this would mean the reverse: we might overestimate the cleaning effect. As the other reviewer pointed out, the underpredictions can also be seen for PNC and LDSA. For that reviewer comment, we provided our best estimate for the reasons for having this underprediction.

As a complementary material for this response, we have provided one illustration on how modelled $PM_{10}$ concentrations compare against station measurements in the Helsinki region, as a function of road surface moisture. There is a clear dependency on road surface moisture and $PM_{10}$ concentrations, as can be seen from the figure below. However, we don't observe a clear tendency to over-/underestimate $PM_{10}$ in certain road moisture conditions.

[Figure]

Figure R1: Complementary illustration of modelled and measured PM10 concentrations in several stations in the Helsinki region in 2024. Measured concentrations have been shown as black narrow bars. The presented

values have been conditioned on road surface water [μm]. "<1.0" corresponds to dry road conditions. The modelling time span is the same as the campaign time span.

Line 470: PM should read PM10

AR: PM is now corrected to $PM_{10}$ in the sentence:

**The modelled $PM_{10}$ concentrations reached their maximum of approximately 75 μg m$^{-3}$ at 10 am, and a minimum value below 20 μg m$^{-3}$ similarly to the measured $PM_{10}$ in the early morning hours.**

Line 521: Do you mean "combustion source" with "burning source"

AR: Yes, this is now changed:

**The most notable underestimation was observed for BC at O10m with wind directions between 300 to 360 degrees (which was the direction of the storage building behind the measurement area), possibly indicating that there was a combustion source in this direction that was not considered by the model.**

Line 575: Do you really mean "significant" in the sense of "statistically significant"? If so, you will need to provide a proof for the significance. Otherwise, I'd suggest to rephrase to "…and lowest for NO2".

AR: The usage of significance in the text is removed from the whole text if no statistical evidence is provided. This is done in response to the Comment 1 from Reviewer 2.

Line 604: I think the "sensors" were actually "samplers" (appears twice in this line).

AR: The word sensors is changed to samplers in this chapter.

**The modelled $NO_2$ average was compared against passive $NO_2$ measurements. According to the results, these were in general agreement with the exceptions in two samplers close to the highway (PAS_6m and PAS_O10m). With these two samplers, a strong overprediction was seen during February and April, but not during May. These discrepancies may have been the result of simplified in-plume $NO_x$-Ozone photochemistry that is being used in Enfuser and meteorological effects affecting the passive samplers. It is also possible that the passive sampler measurements during April were biased indicators for true concentrations.**

**Reviewer 2:**

Manuscript egusphere-2025-1423 by S.D. Harni et al. reports measurements and modelling data during a 3-month measurement campaign in different horizontal distances from a noise barrier. The effect of the noise barrier on particulate pollutants and nitrogen dioxide concentrations is quantified based on measurements and modelled concentrations of a Gaussian air quality model Enfuser that incorporates an obstacle detection and concentration reduction routine for simulating the effect of the noise barrier. An advantage of that model is that it can be informed with complementary datasets of observations, traffic flows and geodata. A caveat of the experimental design is the missing PNC measurement at 10 m and that BC was only measured at 20 m distance behind the noise barrier. The work appears well thought out and executed and forms a coherent study which fits well in the field of air quality research. The manuscript is clearly written but overall lacks transparency in model description and accuracy in data interpretation. I recommend publication if a number of smaller issues addressed below are resolved.

Specific Comments:

> 1.) Abstract: The reduction in pollutant concentrations through the noise barrier are described as being "significant". The word "significantly" is used excessively in the Introduction. The excessive use of "significant" and "significantly" in this manuscript without presenting a check of statistical significance should be avoided. Examples are found on P15, L323; P18, L380, L385, L408, but there are more.

AR: The usage of significantly and significant in the text was reviewed, and the wording was changed because of no statistical evidence to justify the usage of "significant".

> 2.) Introduction (P2, L66-67): CFD models have been used to simulate street canyons and other built environments, the argument of not adopting LES should be more specific, i.e. with respect to fitness for the purpose of this study.

AR: It is true that pollutant concentrations within the measurement campaign area (that is, relatively small) could be simulated with, e.g., LES models. The long measurement campaign duration that covers several months, however, makes the task, in our opinion, unfeasible for LES modelling. Our purpose is to test the feasibility of lighter-weight simulation tools to address potentially every noise barrier in the urban area. Another issue with LES-simulation is that the contributions from emission sources at greater distances would have to be ignored, or utilise some nested approach. As such. The use of LES modelling does not fit the purpose of this study.

> 3.) Modelling framework (P8): I understand the brevity of the modelling framework section, by referring readers to the published model description of the Enfuser model. Nevertheless, it should be possible to understand the model features which are utilized in this study, without having to consult this publication. I suggest adding two pieces of information: a) how many receptors are used for the Helsinki metropolitan area with base resolution and the campaign area of 4x4 m2 (it should be meter square in the text); and b) describe the calculation of pollutant concentrations in the vertical –

are they inferred from the SILAM CTM model? – which is relevant when you compare to the vertical profile of the drone measurements.

AR: We have added the suggested pieces of information in the text right under section 2.4 (Modelling framework):

Enfuser is an urban-scale air quality model (Johansson et al., 2022). The modelling approach for local emission sources is Gaussian (a combination of Gaussian plume and puff methodologies), and long-range transportation of pollutants is addressed by incorporating a regional-scale AQ forecast to define hourly background concentrations. As a novelty, the model uses AQ measurement-driven data assimilation to adjust these background concentrations, but also emission-source specific emission factors on an hourly basis.

Now the reader should have a better understanding of the modelling approach without accessing the model description paper. For brevity, we would like to avoid technical details here; for example, we simply state that we provide model predictions at 2m above ground, but avoid the discussion on how the height measure (verticality) is specifically addressed in the Gaussian approach. We also model pollutant concentrations at the specific measurement heights at the exact sensor / AQ station locations, which we had already included in the description.

The role of the SILAM CTM model is to provide a regional background, which has been assumed to have a uniform vertical distribution in this study. On top of the regional background, we model pollutant contributions within the Helsinki region with a Gaussian plume and puff–based approach. This approach, due to the dual use of plume and puff modelling, is complicated and is difficult to describe in this paper in a satisfactory manner. Thus, we feel best to refer to the model description paper, which is open to everyone.

> 4.) Modelling of noise barrier (P10): the description of the reduction effect of the noise barrier on concentrations at receptors is adequate. Some more details on the precomputed obstacle detection should be given, such as the workflow of this routine and the datasets (topography, building heights, etc.) used. Equation (19) contains a duplicate comma. How is the concentration at the obstacle itself calculated or are the obstacles masked in the 2D concentration map?

AR: The topography is given by NASA SRTM, and building heights are based on OpenStreetMap data. To be precise, most buildings do not have a height parameter in the OSM data, and in such cases, we use 100x100m Global Human Settlement (GHS) averaged building height data. In essence, together with these datasets, we have a precomputed digital surface map (DSM) at our disposal. We also use a land-use mask so that we can easily classify any location of interest as e.g., a building, a road, etc.

Unfortunately, the Gaussian modelling approach is ill-suited to deal with complex terrain and in most cases, we cannot address it adequately. In some simple cases, e.g, an emission source is at an elevated location (a hill), we can modify the height parameter of the Gaussian equations accordingly.

The concentrations within obstacles and buildings can be disregarded as placeholders. In the future, the model may be developed further to address indoor air quality. With the current version, the model simply detects that the raster point is within an obstacle (e.g., a building or

a wall) and applies a statistical reduction (e.g., 80% from local emission source contributions).

In conclusion, we added this information to the text. Also, the duplicate comma has been removed (thank you for spotting that out).

> 5.) LDSA, BC and PNC modelling (P12): Please provide details of the proxy of PNC emission factor based on PM2.5 emission. The PNC emission factors are probably different for various emission sectors. A table with that information would be useful. Is there any size segregation of PNC in the model (the information should be placed here)?

AR: For both LDSA and PNC, the emission proxies based on PM2.5 are indeed different for different emission sectors. We have added a clarification about this in the text.

We could table the values we have used, but the usage of Enfuser data assimilation makes them ultimately freely floating and evolving parameters. For example, in case we put too high an emission factor for LDSA or PNC for e.g., traffic, then the emission factors begin to gradually decrease over time into a value that is in better agreement with measurement evidence. This is one of the novel features of the model that is presented in the model description paper, and is, in our opinion, too long to describe in this paper. Similarly, the background concentration for LDSA and PNC are heavily influenced by the data assimilation. Considering the many measurement stations in the Helsinki region, we get to adjust the hourly background concentration values with high confidence with this approach, regardless of our initial proxy value based on PM2.5.

At this point, we do not address the issue of size segregation of PNC since the measurement network for PNC only provides the flat particle number count. In the future, if size distribution would also be available, we could attempt a more realistic modelling approach for PNC that could, e.g., address coagulation effects for the smallest fraction. We did already mention this in the text, so we left the document unmodified in this regard.

> LDSA represents surface area concentration of particles deposited in the alveolar region of human lungs and depends on the size distribution of particles. The most common size range for particle surface area is in the range of 100–500 nm. Obviously, LDSA is simulated as passive tracer like the other particulate pollutants, with emission as a fraction of PM2.5.

> Please explain which algorithm or post-processing is used to calculate the LDSA concentration field. Assimilation of LDSA and PNC background – again, is this based on available concentration measurements of the respective component?

AR: Technically speaking, the computation of an LDSA concentration field is similar to the computation of $PM_{2.5}$ and PNC concentration fields. The measurement data that drives the data assimilation (affecting hourly emission factors for different emission source sectors) are hourly LDSA values. This means that we do not need to incorporate an algorithm to compute LDSA based on other modelled properties such as particle size distributions. As a side note, even with this simplistic modelling approach, LDSA is one of the pollutant species with the highest correlations when we cross-compare the results against the measurement locations in

Helsinki (not shown in the paper). This goes to show that LDSA can apparently be treated as a passive tracer quite successfully.

As regards the assimilation of the LDSA and PNC background, it is indeed based on measurements, as we have written in the paper. Naturally, there is no "component split" for measurements, so the true background is latent information. It is the task of the data assimilation procedure to fine-tune the background concentrations while simultaneously adjusting emission source factors for local emission sources.

This assimilation process has been described in detail in Johansson et al. (2022) and is, in our opinion, too long a description to be added in this paper. We could describe this as "an optimisation task to minimise the weighted sum of squared prediction errors at measurement locations, while searching for optimal background adjustments and emission factor modifications with a gradient descent search algorithm". Then again, this would probably raise more questions than it answers.

> 6.) P3, L298-300: "The gradients of large particle might also be due to larger particles having slower dispersion" – seems to contradict the findings of the cited study by Zheng et al. (2022), who noted that larger particles (coarse mode, > 1 µm aerodynamic diameter) are more affected by traffic-induced turbulence than smaller particles, which would indicate more efficient dispersion. A reference for the apparently slower dispersion of larger particles should be given here.

AR: Now, two articles showing that smaller particles are more easily diluted with air (Kumar et al., 2008) and that the deposition of the larger particles is faster (Noll et al., 2001) are added to the manuscript and to the reference list:

**Kumar, P., Fennell, P., and Britter, R.: Effect of wind direction and speed on the dispersion of nucleation and accumulation mode particles in an urban street canyon, Sci. Total Environ., 402(1), 82–94. https://doi.org/10.1016/j.scitotenv.2008.04.032, 2008.**

**Noll, K. E., Jackson, M. M., and Oskouie, A. K.: Development of an atmospheric particle dry deposition model, Aerosol Sci. and Technol., 35(2), 627–636. https://doi.org/10.1080/02786820119835, 2001.**

> 7.) Figure 6 and belonging text: PNC should also follow a logarithmic curve, as many studies have demonstrated. Is this not observed because of the missing measurement point at 10 m? At least, the linear dashed line in figure part D appears unrealistic. The NOx measurement data should be displayed as well, since NOx is not affected by the chemical conversion and rather behaves like a passive tracer.

AR: The authors agree that the PNC should follow a logarithmic curve. The authors also argue that this is the case in the case of the 20$^{th}$ of March. However, this is not as clear as for other variables, as the data from the 10m point is indeed missing. The dashed line is a bit more unrealistic looking, as on the 24$^{th}$ of March, the data from the 20m pole was missing. Therefore, for the 24$^{th}$, the line only shows the change of concentrations while not addressing the shape of the gradient. Authors agree that displaying NO$_x$ data could potentially be

beneficial by not being affected by chemical composition. However, as NO$_x$ results have not been presented elsewhere in the manuscript, adding them only to one figure would be confusing.

Additionally, a typo in the caption of Figure 6 was spotted and corrected:

PMC changed to PNC

> 8.) Drone measurements (P16): is the PNC at 15 m height in open area significantly lower than at 2 m height? The labels in the plots of Figure 7 are unclear, they should indicate that 2 m and 15 m are in the vertical.

AR:  The authors agree that the difference between 15m and 2m heights in the open area is minimal, and this is now better stated in the text:

**Also, the lowest concentrations were measured in an open area at an elevation of 15m, although the concentrations were only marginally lower.**

Figure 6 was edited to now also mention that the 2m and 15m refer to height above ground:

[Figure]

**Figure 2: Boxplots of PNC measured with a drone in the open area (O 2m  a.g., O 15m  a.g.) and behind the noise barrier (NB 2m a.g., NB 15m  a.g.) separately for the 20th and 24th of March.  The median values are indicated with horizontal red lines, the blue box resembles the lower and upper 25th and 75th percentiles, the whiskers represent the lowest and highest values considered for data evaluation, and the outliers are marked with red plus marks. The a.g. stands for above ground.**

> 9.) P18, L380-390: Add the percentage contribution of direct traffic emission to the modelled concentrations.

AR: These percentage contributions have been added. We used O20m for this purpose, as it exists for all measured pollutant species. Now the text reads:

"The traffic-related particles had quite noticeable contributions to all BC, LDSA, PNC and PM$_{10}$. For example, at O20m the traffic-related fractions are 70%, 45%, 69% and 75% respectively.

10.) P 20, L420-425: The model also underestimates high PNC peaks, which cannot be explained by the use of studded tyres.

AR: This is certainly true, but the underprediction is most prominent with PM$_{10}$, and the best explanation is an issue with resuspension, snow-cover and studded tires. The PNC (and partly LDSA) underprediction during this time could be explained by improper atmospheric conditions (e.g., stability, inversion) and local wind conditions during wintertime. We also added this speculation in the text. Our best explanation is a mixture of the two possible explanations.

The other reviewer also had a question about this topic (and the effects of precipitation), and we showed selected PM$_{10}$ modelling results at the other measurement locations in the Helsinki region, during the measurement campaign. Referring to this external quality control data at the measurement stations, we, the authors, can observe that there is no clear underprediction of PNC, LDSA and PM$_{10}$ at the measurement stations (outside of the campaign area) during March. This confirms to us that the reasons for these unpredictions are local, and thus are most likely linked to the local road emissions and road conditions.

11.) P23, L475-480: Compare the range of measured and modelled LDSA concentrations of this campaign to other LDSA measurements in Helsinki and other Finnish cities.

AR: Authors agree that this would be a valuable comparison, and a short comparison to LDSA concentrations in Helsinki is added to the text. However, LDSA is still rarely measured as such (although it can be calculated from particle number size distribution data such as SMPS), and only a couple of studies reporting LDSA in Finland were found, and text related to them was added to the manuscript.

**Similar concentrations for traffic environments have also been reported in cities of Helsinki (13.2 – 35.4 μm$^{-2}$ cm$^{-3}$) and Tampere (12.2 – 47.9 μm$^{-2}$ cm$^{-3}$) (Kuula et al., 2020; Lepistö et al., 2023). Similar LDSA concentrations (9.4 μm$^{-2}$ cm$^{-3}$) to the minimum have been observed in urban background areas in Helsinki (Kuula et al., 2020). In Helsinki, in residential areas, the LDSA concentrations are measured between the urban background and traffic environments at 12 – 22.6 μm$^{-2}$ cm$^{-3}$ (Kuula et al., 2020; Lepistö et al., 2023). Similar concentrations of 22.5 μm$^{-2}$ cm$^{-3}$ at residential areas have also been measured at Raahe (Lepistö et al., 2023). Modelled LDSA concentrations have earlier been compared to measured LDSA concentrations in Finland at traffic environment and urban background, with mean absolute errors of 3.7 and 2.3 μm$^{-2}$ cm$^{-3}$, respectively (Fung et al., 2022).**

And also, additional references are added to the reference list:

**Fung, P. L., Zaidan, M. A., Niemi, J. V., Saukko, E., Timonen, H., Kousa, A., Kuula, J., Rönkkö, T., Karppinen, A., Tarkoma, S., Kulmala, M., Petäjä, T., and Hussein, T.: Input-adaptive linear mixed-effects model for estimating alveolar lung-deposited**

surface area (LDSA) using multipollutant datasets. Atmos. Chem.  Phys., 22(3), 1861–1882. https://doi.org/10.5194/acp-22-1861-2022, 2022.

Lepistö, T., Lintusaari, H., Oudin, A., Barreira, L. M. F., Niemi, J. V., Karjalainen, P., Salo, L., Silvonen, V., Markkula, L., Hoivala, J., Marjanen, P., Martikainen, S., Aurela, M., Reyes, F. R., Oyola, P., Kuuluvainen, H., Manninen, H. E., Schins, R. P. F., Vojtisek-Lom, M., Ondracek, J., Topinka, J., Timonen, H., Jalava, P., Saarikoski, S., and Rönkkö, T: Particle lung deposited surface area (LDSAal) size distributions in different urban environments and geographical regions: Towards understanding of the PM2.5 dose–response. Environ. Int., 180. https://doi.org/10.1016/j.envint.2023.108224, 2023.

12.) Figure 11: it is very confusing for the reader that NB20m and O10m plots of BC are placed next to each other, because the comparison does not reveal the effect of the noise barrier – as it is the case for PM10, LDSA and PNC in the same figure. My suggestion is to either place a note of caution about this in the caption or even better, to move the BC plots to an Appendix figure.

AR: In this paper, we present several other illustrations (e.g., Figs. 6 and 8) that are better suited for revealing the effects of the noise barrier for BC. In this figure, our objective is to provide maximum contrast ('NB' on the left, vs 'O' on the right), and to present emission source contributions as a function of wind direction. Respectfully, we still feel that 'O10m' for BC is the better candidate for this figure than 'O20m'. We kindly propose that we keep the figure as it is.

13.) Conclusions: regarding the simulation of the noise barrier (P29, L589-601), it should be discussed that the reduction effect of the barrier might be different for particles than for gases, as particles may deposit on the vertical surfaces of the barrier.

AR: We agree that the reduction effect might be different for particles, and potentially these are different depending on the particle size category as well. In this study, we didn't feel comfortable adding yet another hyperparameter, so we ignored this possibility. We have now mentioned this, but in Section 2.4.1 instead of Conclusions.

Technical corrections:

P19, L412: in the unit of LDSA, delete space in cm and add a space between μm and cm.

AR: The sentence is now corrected:

For LDSA, the observed concentrations behind the barrier and in the open area were at similar levels, being only slightly lower behind the noise barrier with a daily average reaching up to 23 $\mu m^{-2}$ $cm^{-3}$.

P23, L470: replace "PM" by "PM10".

AR: Done

The modelled $PM_{10}$ concentrations reached their maximum of approximately 75 $\mu g$ $m^{-3}$ at 10 am, and a minimum value below 20 $\mu g$ $m^{-3}$ similarly to the measured $PM_{10}$ in the early morning hours.

P17, L359-362: change to present tense. Same for P18, L392-393.

AR: This is done for both texts. Lines 359-362:

**In the next sections, we present various results where modelled and measured pollutant concentrations are compared. The modelled and observed average pollutant concentrations for BC, PM$_{10}$, LDSA, and PNC are shown in Fig. 8 for the 6 or 4 measurement poles, depending on the pollutant species, over the whole measurement campaign period. The split between emission source categories is shown for the model predictions in the form of staggered columns.**

And lines 392-393:

**The measured and modelled pollutant concentrations for the main measured pollutants BC, PM$_{10}$, LDSA, PNC, and NO$_2$ are compared in Fig. 9 in the form of daily averages.**

Additional alterations made to the text:

While making the changes proposed by the reviewers, some small mistakes in the text, such as missing commas, double citations and mispelled words throughout the text were corrected.